# Effect of Medium Chain Triglycerides on the Digestion and Quality Characteristics of Tea Polyphenols-Fortified Cooked Rice

**DOI:** 10.3390/foods12234366

**Published:** 2023-12-04

**Authors:** Ying Li, Liya Niu, Chao Sun, Dongming Li, Zicong Zeng, Jianhui Xiao

**Affiliations:** 1School of Food Science and Engineering, Jiangxi Agricultural University, 1101 Zhimin Road, Nanchang 330045, China; ly18370625871@163.com (Y.L.);; 2Key Laboratory of Crop Physiology, Ecology, and Genetic Breeding, Ministry of Education, Jiangxi Agricultural University, Nanchang 330045, China

**Keywords:** medium chain triglycerides, starch-lipid complexes, tea polyphenols-fortified cooked rice, digestion characteristics

## Abstract

Nowadays, medium chain triglycerides (MCT) with special health benefits have been increasingly applied for fortifying food products. Therefore, the present work aimed to investigate the effects of MCT on traditional tea polyphenols-fortified cooked rice (TP-FCR). It was visualized by DSC, CLSM, XRD, FT-IR, and Raman spectroscopy. The higher content of starch-MCT complexes with an increase in the relative crystallinity and the generation of short-range ordered structures contributed to a more ordered and compact molecular arrangement, which can hinder the action of digestive enzymes on starch. SEM demonstrated that MCT transformed the microstructure of TP-FCR into a denser and firmer character, making it an essential component hindering the accessibility of digestive enzymes to starch granules and slowing the release of tea polyphenols in TP-FCR to attenuate starch digestion. Consequently, the addition of MCT reduced the polyphenol-regulated starch digestibility from 74.28% in cooked white rice to 64.43% in TP-FCR, and further down to 50.82%. Besides, MCT also reduced the adhesiveness and improved the whiteness of TP-FCR. The findings suggested that MCT incorporation could be a potential strategy in cooked rice production to achieve high sensory quality and low glycemic cooked rice.

## 1. Introduction

Tea products rich in polyphenols have functional properties such as hypoglycemia, antioxidants, obesity prevention, etc. [1]. Adding tea products to produce tea polyphenols-fortified foods that integrate multiple functions and nutrients is very popular among consumers. Tea polyphenols-fortified cooked rice (TP-FCR), which is prepared by cooking rice with tea products, is traditionally one of the most common staple foods in many Southeast Asia countries ascribed to their unique flavor. Furthermore, many studies have found that tea polyphenols can interact with starch to inhibit the digestion of starch [2,3,4]. Also, our research group demonstrated that instant green tea could enhance the palatability and slow the in vitro digestion properties of cooked rice [5].

To slow starch digestion, besides the interaction between tea polyphenols (TP) and starch, the interaction between lipids and starch is also important. Ai et al. proposed that lipids can interact with starch to reduce the sensitivity of starch to enzymatic hydrolysis and improve the anti-digestibility of starch [6]. Luangsakul and Ritudomphol added oil to rice for cooking and revealed that the formation of amylose-lipid complexes in cooked rice reduced in vitro starch digestibility and predicted glycemic index [7].

What is more, many studies have proven that the type of lipid and its chain length and degree of unsaturation can affect complex formation, structure, and functional properties. Sun et al. proposed that maize starch interacting with different types of fatty acids (FAs) exhibited different V-type polymorphs, and the FAs with a chain length of 10–14 carbons showed stronger intermolecular interactions with MS and produced more ordered structures than other FAs [8]. Zabar et al. demonstrated that the crystallinity and thermostability of amylose-long chain fatty acid complexes decreased with increasing degree of unsaturation of FAs [9]. Therefore, the selection of lipids with appropriate chain length and high saturation contributes to the formation of more stable starch-lipid complexes.

Medium chain triglycerides (MCT) have high saturation and shorter carbon chain, are colorless and odorless at room temperature, and have low viscosity and good stability [10,11], which facilitate the formation of complexes with starch. Additionally, MCT possesses functional properties such as obesity prevention and improvement of carbohydrate, lipid, and cholesterol metabolism [12,13]. Nowadays, there are various foods enriched with MCT available in the market. Consequently, MCT with good physicochemical and functional properties was selected as a quality modifier, which was expected to further enhance the nutritional value and eating quality of TP-FCR.

In addition to the interaction between two components, such as starch and polyphenols, as well as starch and lipids, there are also three component interactions in the food system. Wang et al. reported that starch can interact with β-lactoglobulin and fatty acid (FA) to form ternary complexes, which had a larger short-range molecular order and higher relative crystallinity than binary starch-FA complexes [14]. Lin et al. suggested that proteins promoted the formation of starch-protein-linoleic acid (LA) complexes due to its emulsification, which had more ordered structures and lower starch digestibility than the LA- and proteins-complexed starches [15]. Starch, lipids, and protein are the three macronutrients in many food systems and their interactions have been extensively studied. However, the interactions between starch, lipids, and polyphenols and the applications of these interactions in foods have been less studied. Whether MCT can interact with the polyphenols and starch in TP-FCR to improve its quality is also unclear.

Therefore, this study aimed to investigate the effects of MCT on the digestive properties and quality of TP-FCR. The improvement of MCT on TP-FCR was evaluated by surface porosity, texture analysis, and color. Meanwhile, differential scanning calorimetry (DSC), X-ray diffraction (XRD), Fourier-transform infrared spectroscopy (FT-IR), Raman spectroscopy, confocal laser scanning microscope (CLSM), and scanning electron microscopy (SEM) were used to interpret the effects of MCT on TP-FCR in terms of the multi-scale structure. The effects of MCT on the digestive properties of TP-FCR were also characterized by dynamic modulus changes and fluorescence quenching spectrum during digestion.

## 2. Materials and Methods

### 2.1. Materials

Golden Dragon Fish Northeast Pearl Rice (japonica rice) with 15.8% amylose content was purchased from Yihai Kerry Arawana Holdings Co., Ltd. (Jiamusi, China). MCT was purchased from Shanghai HeYi Food Technology Co., Ltd. (Shanghai, China). The fatty acid composition of MCT is caprylic acid glyceride and decanoic acid glycerol ester, and the purity of MCT is 100%. Instant green tea powder (IGT) was purchased from Jiangsu Dehe Biotech Co., Ltd. (Wuxi, China). IGT is a water-soluble component extracted from green tea leaves, mainly consisting of tea polyphenols, flavonoids, alkaloids, theanine, and tea polysaccharides, with a tea polyphenol content of 18.36%, and free of carriers and other additives such as saccharides. Porcine pancreatic amylase (PPA) was purchased from Aladdin Reagent Co., Ltd. (Shanghai, China). DNS reagents and aspergillus niger amyloglucosidase (AAG 100 U/g) were purchased from Solarbio Technology Co., Ltd. (Beijing, China). All other chemical solvents were analytical grade and purchased from Sinopharm Chemical Reagent Factory (Shanghai, China).

### 2.2. Cooked Rice Samples Preparation

A total of 100 g of rice was rinsed three times and filtered; 150 mL of deionized water dissolved with 2 g of IGT was poured into the rinsed rice in a rice cooker (SF12FB627, SUPOR Co., Ltd., Hangzhou, China). Then, different amounts of MCT (0, 1, 2, 3, 4, and 5 g) were added respectively and mixed well. Next, the rice was soaked for 30 min, cooked, and kept warm for 20 min. Subsequently, TP-FCR samples separated from the middle of the cooker were put into plastic bags with a moisture vapor transmission rate of approximately 0.84 mg/(m^2^·h) to prevent moisture loss and cooled to room temperature.

At the same time, some of the TP-FCR samples were taken out and frozen at −80 °C for 48 h and lyophilized with the freeze-dryer (LC-10N-50D, Shanghai anting scientific instrument Co., Ltd., Shanghai, China) at a vacuum degree of 25 Pa and a temperature of −45 °C for 48 h. Each lyophilized TP-FCR sample was divided into two portions, one of which was left intact, and the other was ground into powder for further detection. The TP-FCR without MCT was named as control.

### 2.3. Surface Porosity

A whole pot of TP-FCR was placed on the Perfection V330 Photo Scanner (EPSON Scan, Epson (China) Co., Ltd., Beijing, China) without damaging the shape and then the image of the TP-FCR sample was taken. The surface porosity of TP-FCR was calculated using Image j 1.48u software [5].

### 2.4. Texture Profile Analysis (TPA)

Texture properties of TP-FCR samples were tested using a texture analyzer (TA-XT2i, Stable Microsystems Co., Ltd., Surrey, UK) with cylindrical probe P/0.5, according to a modified method [16]. TP-FCR samples separated from the middle of the rice cooker were cooled at room temperature for 30 min and immediately placed on the test bench of the texture analyzer for testing. Each sample with a 50% compression ratio was tested 10 times. The pre-test and post-test speed was 2 mm/s and the interval time between two compressions was 5 s. Parameters, such as adhesiveness, hardness, and chewiness were obtained using the TPA test. Adhesiveness represents the force required to peel off the food when it is attached to people’s tongue, teeth, mouth, etc. [17], hardness represents the force required for teeth to deform food, chewiness indicates the energy required to chew solid food until it could be swallowed [18].

### 2.5. Color

The color of TP-FCR samples was measured using a Hunter-Lab colorimeter (Color Quest XE, Hunter Associates Laboratory, Fairfax, VA, USA), and the results were expressed as the values of lightness (L*), redness/greenness (a*), and yellowness/blueness (b*) [19]. The whiteness (W) of TP-FCR was calculated according to the following formula:$$W={100-[{\left(100-L^{*}\right)}^{2}+a^{*2}+b^{*2}]}^{1/2}$$

### 2.6. Water Absorption Rate

The water absorption rate was determined according to the method of Wei et al. with slight modifications [20], TP-FCR samples prepared in Section 2.2 were weighed and the water absorption rate was calculated according to the following formula:$$Water\;absorption\;rate=\frac{\left(W-100\right)}{100}$$
where W is the weight of 100 g of rice after cooking.

### 2.7. Swelling Ratio

According to the method reported by Paesani and Gómez with some modifications [16], the drainage method was used to calculate the swelling ratio of TP-FCR. Briefly, 100 g of rice before cooking was measured with a measuring cylinder containing 100 mL of water, and then the volume of TP-FCR samples (10 g) was measured in the same way. The swelling ratio of TP-FCR samples was calculated as follows:$$Swelling\;ratio=\frac{\left(\frac{V_{2}}{10}\times W-V_{1}\right)}{V_{1}}$$
where V_1_ is the volume of rice (100 g) before cooking, V_2_ is the volume of the TP-FCR samples (10 g), and W is the weight of 100 g of rice after cooking.

### 2.8. Differential Scanning Calorimetry (DSC)

Thermodynamic properties of TP-FCR were determined using differential scanning calorimetry (DSC 214 Polyma, NETZSCH, Ahlden, Germany). A total of 8 mg of the lyophilized powder of the sample was accurately weighed into an aluminum crucible, then sealed and balanced for 12 h before testing [21]. An empty aluminum crucible was used as a control and the crucible temperature was increased from 20 °C to 170 °C at a rate of 10 °C/min. The starting temperature (T_o)_, the peak temperature (T_p_), the termination temperature (T_c_) and the enthalpy change (ΔH) were recorded.

### 2.9. X-ray Diffraction (XRD)

Referring to the method described by Seo et al. [22], the XRD spectroscopy of lyophilized powder of TP-FCR was obtained by using an X-ray diffractometer (XRD-7000; Shimadzu Corp., Kyoto, Japan) with a scanning voltage of 40 kV and a scanning current of 30 mA. The scan rate is 2°/min with the scan range 2θ of 4–40° at a scan step of 0.02°.

### 2.10. Fourier-Transform Infrared Spectroscopy (FT-IR)

According to the method of Lv et al. [23], the lyophilized powder of TP-FCR samples was ground with KBr and then tested using an FT-IR spectrometer (Nicolet iS5, Thermo Fisher Scientific Co., Ltd., Waltham, MA, USA) at the scanning range of 400–4000 cm^−1^. The scanning resolution at 4 cm^−1^ and the cumulative number of scans at 32 were set.

### 2.11. Raman Spectroscopy

According to a modified method of Sivam et al. [24], the lyophilized powder of TP-FCR was placed on a slide, flattened, and tested using a Raman spectrometer (DXR2, Thermo Fisher Scientific Co., Ltd., Waltham, MA, USA) with a 785 nm semiconductor laser light source. The exposure time and the slit were set at 10 s and 25 μm, respectively. Each sample was scanned 10 times in the laser range of 200–3200 cm^−1^.

### 2.12. Confocal Laser Scanning Microscope (CLSM)

The lyophilized powder of TP-FCR was mixed with distilled water in the ratio of 1:7 and then 20 μL fluorescein isothiocyanate (FITC, 0.1%, *w*/*v*) was added to stain the starch. And 20 μL Nile Red (0.1%, *w*/*v*) was added to stain the MCT [25]. The samples were mixed evenly and kept away from light for 2 h, and then observed by the confocal laser scanning microscopy (Leica TCS SP5, Leica Microsystems GmbH, Wetzlar, Germany).

### 2.13. Scanning Electron Microscopy (SEM)

Based on the method described by Lu et al. with appropriate modifications [26], the microstructures of the intact lyophilized TP-FCR cross-section and surface were observed by scanning electron microscopy (Zeiss evo 18; Carl Zeiss Co., Ltd., Oberkochen, Germany) at 100× and 500×.

### 2.14. Viscoelastic Properties during Digestion

The dynamic modulus change during digestion was determined by a rheometer (Discovery HR-1, TA Instrument Co., Ltd., New Castle, DE, USA) with reference to the method described by Xiao and Zhong with slight modifications [27]. The concentric cylinder was selected for oscillation time scanning. The TP-FCR samples and sodium acetate buffer were crushed into homogenate by a juicer (JYL-C020E, Joyoung Co., Ltd., Nanjing, China) at a ratio of 1:1. The digestive enzyme solution was placed in a shaking water bath (SHA-C, Changzhou Guoyu Instrument Manufacturing Co., Ltd., Changzhou, China) at 37 °C until the temperature of the digestive enzyme solution reached 37 °C to activate the digestive enzymes. Then the above homogenate was mixed with the pre-activated digestive enzyme solution (PPA:AAG = 20:1, *w*/*w*) at a ratio of 9:1, and added to the concentric cylinder for testing. The program scan time was 3 h, the frequency was 1 Hz, and the temperature was set at 37 °C. The samples without enzyme were used as control and the whole procedure was performed in the linear viscoelastic zone.

### 2.15. Fluorescence Quenching during Digestion

A 0.2 g of TP-FCR lyophilized powder was added to sodium acetate buffer (pH 5.2), and the pH was adjusted to 6.5 with 1 mol/L and 0.1 mol/L NaOH, then the pre-activated mixed enzyme solution (PPA:AAG = 20:1, *w*/*w*) was added and maintained in a shaking water bath at 37 °C for 3 h. Fluorescence spectra of the supernatant of TP-FCR samples digested for 0, 20, 40, 60, 90, 120, 150, and 180 min were performed with a fluorescence spectrophotometer (970CRT, Shanghai Precision and Scientific Instrument Corp., Shanghai, China) by referring to the method of Chen et al. [28]. Both excitation and emission slits were set to 5 nm, and the fluorescence intensity of samples in the emission wavelength range of 250–550 nm were recorded at the excitation wavelength of 280 nm.

### 2.16. In Vitro Starch Digestion

For the in vitro digestion of cooked white rice and TP-FCR samples, the test method of Fu et al. with slight modification was referred to [29]. A 0.8 g of lyophilized powder of TP-FCR was mixed with sodium acetate buffer (pH 5.2), 1 mol/L and 0.1 mol/L NaOH to make the pH of the system at 6.5. 5 mL of pre-activated enzyme mixture (PPA:AAG = 20:1, *w*/*w*) was added and maintained in a shaking water bath at 37 °C for 0, 20, 40, 60, 90, 120, 150, 180 min, respectively. Anhydrous ethanol was then added to inactivate the enzyme. The mixture was then centrifuged and separated the supernatant. The glucose content at different times was measured using the DNS method and the contents of rapidly digestible starch (RDS), slowly digestible starch (SDS), and resistant starch (RS) were calculated based on the glucose content. The RDS, SDS, and RS contents were calculated according to the following formulas:$$\mathrm{RDS}(\%)=\frac{\left(G_{20}-\mathrm{FG}\right)\times 0.9}{\mathrm{TS}}\times 100$$
$$\mathrm{SDS}\left(\%\right)=\frac{\left(G_{120}-G_{20}\right)\times 0.9}{\mathrm{TS}}\times 100$$
$$\mathrm{RS}\left(\%\right)=\frac{\mathrm{TS}-\left(\mathrm{RDS}+\mathrm{SDS}\right)}{\mathrm{TS}}\times 100$$
where G_20_ and G_120_ are the glucose content released within 20 min and 120 min, respectively, FG is the content of free glucose before enzymatic hydrolysis, and TS is total starch content.

The digestion rate constant (k) was obtained by fitting the digestion rate curve of starch. The starch digestion data have often been fitted to a first-order equation [30]:$$C_{t}=C_{\infty }\left(1-e^{-\mathrm{kt}}\right)$$
where C_t_ is the percentage of digested starch at a given time, C_∞_ is the estimated percentage of starch digested at the end of the reaction, “t” is the digestion time, and “k” is the digestion rate constant.

The hydrolysis index (HI) and predicted glycemic index (pGI) were calculated using the following formulas [31]:$$\mathrm{HI}/\%=\frac{{\mathrm{AUC}}_{\mathrm{sample}}}{{\mathrm{AUC}}_{\mathrm{fresh}\;\mathrm{white}\;\mathrm{bread}\;}}\times 100$$
pGI=8.198+0.862×HI
where AUC_sample_ is the area under the hydrolysis curve of sample and AUC_fresh white bread_ is the area under the hydrolysis curve of the reference sample (fresh white bread).

### 2.17. Statistical Analysis

All test data were obtained by at least three tests and expressed as the mean ± standard deviation. The data was analyzed by the one-way ANOVA and Duncan’s multiple range test at a significance level of 0.05 using SPSS 20.0 software, and all figures were drawn using Origin 2021.

## 3. Results and Discussion

### 3.1. Surface Porosity, Texture Properties, and Color

The results of the surface porosity in Figure 1A demonstrated that the addition of MCT was positively correlated with surface porosity, and the more MCT was added, the greater the surface porosity was. This indicated that the cooked rice became fluffy and porous, and had low adhesion between the TP-FCR grains after the addition of MCT. The adhesiveness index in texture properties also reflected the adhesion of TP-FCR, as shown in Figure 1B, the adhesiveness of TP-FCR decreased with the increase of MCT, which was consistent with the results of surface porosity. MCT has good lubricity, which can play a lubricating role by covering the surface of starch or forming starch-MCT complexes to wrap on the surface of starch granules in the form of an insoluble film [32]. In addition, it can inhibit the hydration and swelling of the starch granules and reduce soluble components on the surface of TP-FCR, leading to a decrease in adhesiveness [33,34]. Therefore, the addition of MCT can make TP-FCR have better dispersibility and distinct particles and inhibit adhesion to each other and to the bottom of the pot.

The TPA experiment, also known as the twice-chewed test, mainly involves compressing the sample twice by simulating the chewing motion of the human mouth [17]. The texture properties of TP-FCR are an important factor in evaluating its edible quality. At present, many scholars have used TPA experiments to study the edible quality of food and analyze the correlation between sensory evaluation and texture properties [19,35]. Sensory evaluation of TP-FCR with different additions of MCT was also conducted in this study, and the scoring criteria are shown in Appendix A. The results (Appendix A) showed that the sensory evaluation results were consistent with the instrumental measurement results. The effect of MCT on the texture of TP-FCR was characterized in Figure 1B–D, it was clear that the more MCT was added, the greater the hardness and chewiness were. This was due to that during heating, MCT can interact with starch to form starch-MCT complexes, and the formation of the complexes reduced the water permeability of the starch granules to retard water absorption and decrease the swelling of the starch granules [36,37]. Besides, the internal pores of TP-FCR became smaller and formed a tighter structure, and these structural changes can be observed in the SEM image of TP-FCR, increasing the hardness and chewiness of the TP-FCR.

Color is an important indicator for evaluating the quality of TP-FCR. The increase in whiteness and brightness of the cooked rice is critical for higher consumer acceptance [19]. It was observed from Figure 1E that the L* values representing lightness and W values representing whiteness were increased with the addition of MCT. The more MCT was added, the greater the L* and W were, which indicated that the whiteness and brightness of TP-FCR were dose-dependent on MCT. The results were consistent with sensory evaluation results. This suggested that MCT can improve the color of TP-FCR.

### 3.2. Water Absorption Rate and Swelling Ratio

It has been reported that the starch can interact with other substances during gelatinization, to affect the water absorption, dispersion, expansion, and solubility of starch [5]. Gerits et al. revealed that the formation of starch-lipid complexes delayed the water absorption of grains, inhibited the swelling of starch, and increased the hardness of starch grains [34]. As shown in Figure 2, the water absorption and swelling rate of TP-FCR decreased with the increase of MCT. This is because the starch-lipid complexes and MCT can form an insoluble film on the surface of starch granules, thereby delaying the time for water entering to the starch granules and reducing the granule swelling [32,33,34]. It was reported that the texture of cooked rice was closely related to its water absorption and swelling rate, and the hardness was higher for cooked rice with lower water absorption and swelling rate [38]. Similarly, in this work, TP-FCR with lower water absorption and swelling rate also had higher hardness.

### 3.3. Thermodynamic Properties

To investigate the interactions between MCT and TP-FCR, the thermodynamic properties of TP-FCR were determined using DSC. As shown in Figure 3A, the DSC spectra of cooked rice (CR), IGT, and all TP-FCR samples showed two peaks in the range of 35.25–73.93 °C and 79.92–159.07 °C, respectively, while MCT showed no endothermic peak in this temperature range. Peak I in Figure 3A reflected the melting of incomplete and low-ordered crystal structures, and peak II was attributed to the melting of the complexes formed by starch with MCT and TP [39,40]. The appearance of peak II in the TP-FCR without MCT could be due to the melting of the starch-endogenous lipid complexes and starch-TP complexes in the TP-FCR [41,42]. As can be seen in Figure 3A_1_,A_2_, the peak area and peak temperature of peak I of TP-FCR decreased with the increase of MCT, while the peak area and peak temperature of peak II increased with the increase of MCT. This indicated that the dissociation enthalpy of incomplete and low-ordered crystal structures decreased with increasing MCT, while the dissociation enthalpy of the complexes increased with increasing MCT, which was consistent with the results in Table 1.

The melting temperatures (T_o_, T_p_, and T_c_) reflected the thermal stability of the microcrystals, while the ΔH value was related to the number of crystals (double helix or single helix structure) [41,43]. It can be concluded from Table 1 that the addition of MCT decreased the dissociation enthalpy, T_p_ values, and the heat absorption peaks range (T_o_–T_c_) of peak I, while increasing the dissociation enthalpy, T_p_ values, and the heat absorption peaks range of peak II. Jakobek proposed hydrophobic interactions and hydrogen bond formation between polyphenols and carbohydrates or lipids [44]. The interaction between polyphenols and starch can form two types of complexes, V-type inclusion and non-inclusion [4]. The hydrophobic interactions enabled the polyphenols to be incorporated into the lipid fraction with a sort of micellar protection [45]. The decrease in the enthalpy and T_p_ value of peak I may be due to the interaction between starch, polyphenols, and MCT, which reduced the incomplete and low-ordered crystal structure in starch, and thus the energy required for microcrystalline melting was decreased. The increase in the dissociation enthalpy and T_p_ value of peak II can be attributed to the formation of more starch-MCT and starch-TP complexes, resulting in more stable crystals in TP-FCR that required more energy to dissociate. The formation of complexes with ordered structures can resist the hydrolysis of digestive enzymes, and the steric hindrance of MCT may reduce the accessibility of digestive enzymes to starch in TP-FCR, which was consistent with the results of in vitro digestion of starch.

### 3.4. XRD

The native rice starches displayed an A-type crystalline structure with diffraction peaks at 15°, 17°, 18°, and 23° 2θ, whereas V-type complexes generally showed diffraction peaks at 13° and 20° 2θ [39,46]. Lipids can enter the hydrophobic cavity of the amylose helices through hydrophobic interactions to form a V-type single helix inclusion [8,47]. Similar to starch-lipid inclusion complexes, starch-polyphenol inclusion complexes also had a V-type crystallinity [4,47]. It can be seen from Figure 4 that the peaks representing A-type crystallization almost disappeared due to the disintegration of the ordered structure and the destruction of the crystalline structure in TP-FCR during the cooking process. It was also observed that all diffractograms showed V-type polycrystalline with a diffraction peak at 20° 2θ, which reflected the interaction of starch with MCT and TP to form V-type single helix inclusion complexes. Relative crystallinity represented the crystal integrity of the starch crystalline region and was highly correlated with digestibility [48]. As illustrated in Table 2, the addition of MCT increased the relative crystallinity of starch in TP-FCR. Since the increase of MCT was added, more MCT and TP interacted with starch to form V-type inclusion complexes, which led to a more ordered and compact molecular arrangement. The high degree of structure order was beneficial to enhance the resistance of starch to enzymatic hydrolysis [8,41].

### 3.5. FT-IR

FT-IR spectrograms (Figure 5) revealed that all samples showed broad peak in the range of 3500–3200 cm^−1^, which was due to the stretching vibration of O-H in TP and starch. The broad peak intensity of TP-FCR at 3500–3200 cm^–1^ decreased with the increase of MCT, which was caused by the increase of MCT and the protective effect of MCT on TP that resulted in more MCT and linear hydrophobic chains of TP being encapsulated in the helical lumen of the starch through hydrophobic interactions and thus interfered with the formation of hydrogen bonds between starch molecules. Similar conclusions that the intensity of broad peaks of starch-TP complexes at 3500–3200 cm^−1^ decreased with the increasing addition of TP were reported by Li et al. [49].

The IR-ratios of (1045/1020) cm^−1^ and (1020/995) cm^−1^ can be used to reflect the short-range orderliness and the helical structure of starch [50,51]. As can be seen in Figure 5, the infrared spectrogram of TP-FCR with MCT showed two different peaks at 1740 cm^−1^ and 2850 cm^−1^ compared with the control TP-FCR. The absorption peak at 1740 cm^−1^ was the C=O stretching vibration peak of MCT and the peak at 2850 cm^−1^ was the stretching vibration peak of -CH_2_ of MCT [8,52]. As shown in Table 2, the ratio of (1045/1020) cm^−1^ increased as the amount of MCT increased, indicating that the structure of TP-FCR was more short-range ordered after the addition of MCT, because the larger the ratio was, the more short-range ordered. Starch interacted with MCT and TP via hydrophobic interactions to form V-type complexes, so that the amorphous amylose and the amorphous long side chain of amylopectin that did not form a double helix could form a single helix structure, thus increasing the degree of short-range order of molecules. The addition of MCT reduced the ratio of (1020/995) cm^−1^, suggesting that MCT interacted with amylose and affected the structure of the amorphous region.

### 3.6. Raman Spectroscopy

The half peak width (FWHM) of the Raman spectrum at 480 cm^–1^ is sensitive to changes in the short-range molecular order of starch, and the FWHM at this location is often used to characterize the degree of starch polymerization [53,54]. A smaller FWHM indicated a more ordered structure of starch short-range molecules, and conversely, a larger FWHM indicated that fewer ordered structures were formed [41,55]. As shown in Figure 6A,B, the FWHM of the Raman spectrum at 480 cm^−1^ decreased with increasing MCT, indicating an increase in the short-range ordering of starch in TP-FCR, which was consistent with the FTIR results. As shown in Figure 6C–H, the characteristic bands from 800 to 1500 cm^−1^ of the spectrum were fitted to the split peaks, and the peak areas at 1130 cm^−1^ and 1340 cm^−1^ were recorded. The characteristic band of 800 to 1500 cm^−1^ was mainly generated by the vibration of glucose monomer, and there were many heterogeneous peaks in this region, which were formed by the accumulation of functional groups and the high overlap of chemical bonds [56]. Among them, the peaks at 1130 cm^−1^ and 1340 cm^−1^ were closely related to the single helix structure of the starch [57]. As shown in Table 2, the area of the Raman characteristic peaks at 1130 cm^−1^ and 1340 cm^−1^ increased with the increase of MCT addition, which indicated that the double helix would be deconvoluted into single-helix structure when MCT and TP were compounded with starch, increasing the single-helix structure.

### 3.7. CLSM

Single fluorescent labeling of CLSM and multi-channel overlay images were used to observe the binding of starch and MCT in TP-FCR. Among them, the starch in TP-FCR showed a green color when stained with FITC, MCT appeared red in color when stained with Nile Red, and the yellow fluorescence fraction represented the binding matrix formed by starch and lipids [25,46,58]. As shown in Figure 7, all labeled green, red, and yellow fluorescence areas increased with increasing MCT. The yellow fluorescence area in Figure 7A_3_ was caused by the interaction of starch with endogenous lipids in TP-FCR to form starch-lipid complexes, where some tiny red dots represented the unbound lipids present in the TP-FCR. The increase in the yellow fluorescence area in Figure 7B_3_–F_3_ suggested that more starch-MCT complexes were formed with increasing MCT. The small red dots appearing in Figure 7B_3_–F_3_ indicate that the MCT attached to the starch surface and the large red dots may be free MCT. It has been reported that the increased resistance of starch to enzymatic hydrolysis through the addition of lipids was directly related to the formation and structure of starch-lipid complexes [37,59] and the increase in the amount of starch-MCT complexes can decrease the digestibility of TP-FCR.

### 3.8. Microstructure of Tea Polyphenol-Fortified Cooked Rice

As shown in Figure 8, the microstructure of TP-FCR with different addition of MCT has significant differences. We can observe from Figure 8A_1_,A_3_ that the cross-section and surface of the control TP-FCR at 100× exhibited a flat structure, which had small, dense pores and thin pore walls. The cross-section of the control TP-FCR at 500× showed a uniform and dense three-dimensional network structure. However, as can be seen from Figure 8B_1_–F_1_,B_2_–F_2_,B_3_–F_3_,B_4_–F_4_, the number and density of pores in the cross-section and the surface of TP-FCR decreased gradually with the increasing addition of MCT, and compared with the control TP-FCR, the cross-section of TP-FCR with MCT was uneven and the pore walls thickened. When the addition of MCT was 4% and 5%, the cross-sectional layers of TP-FCR appeared a densely arranged structure.

Li et al. revealed that hydroxyl groups in TP can interact with water to retain a large number of water molecules [49], and at the same time, TP can interact with starch through non-covalent interactions such as hydrophobic interactions and hydrogen bonding, hindering the interactions between starch chains. In this work, TP-FCR formed a uniform and dense network structure after lyophilized, which was an open structure and easily attacked by digestive enzymes. Zhu et al. proposed that the melting of the crystal structure and the rapid expansion of the granules during gelatinization gave the cooked rice grains a huge expansion force, and the strong migration of water tended to disintegrate and break the grain structure quickly, loosening the cooked rice structure [60]. It has been proposed that amylose and non-starch polymers interact to restrict the entry of water into the granules and inhibit the swelling of the starch, thereby limiting the destruction of the starch granules [61]. TP-FCR with MCT had a denser structure compared with TP-FCR without MCT, which was attributed to the fact that the starch-MCT complexes and free MCT attaching to the surface of the grains changed the water permeation channels and delayed the water migration, thus reducing the swelling of the grains and inhibiting the disintegration of the grain structure.

The control TP-FCR had a uniform and dense network structure as well as thin pore walls, which made it more accessible to digestive enzymes and thus more susceptible to hydrolysis. The formation of starch-MCT complexes caused it to form a dense and compact structure, providing a physical barrier for the diffusion of digestive enzymes into the starch granules and slowing the release of tea polyphenols in TP-FCR, which was conducive to improving the anti-digestibility of TP-FCR.

### 3.9. Viscoelastic Properties during Digestion

Dynamic modulus changes during digestion can reflect the rate of TP-FCR digestion. As can be seen in Figure 9A–F, the storage modulus of all TP-FCR samples without enzyme increased with time, which was attributed to the short-term aging of starch, while the loss modulus did not change significantly. In contrast, the storage and loss modulus of all samples with the addition of digestive enzymes in the first 20 min of digestion decreased at a faster rate and then decreased slowly until they stabilized. This was due to the rapid hydrolysis of starch by digestive enzymes at the initial stage, which led to the destruction of the formed network structure and the decrease in the size and concentration of the effective molecules in the system [62,63,64]. At the same time, it can be seen from Figure 9A–F that the storage and loss modulus of TP-FCR without MCT decreased more significantly under the action of enzymes than that of the TP-FCR with MCT. Moreover, higher MCT resulted in a slower decrease. The reduction in the rate of decrease of storage and loss modulus indicated that MCT could delay the enzymatic hydrolysis of TP-FCR and inhibit the digestion of TP-FCR. The inhibition of TP-FCR digestion may also be related to the inhibitory effect of TP on digestive enzymes in addition to its structural effects; therefore, the conjecture was verified using fluorescence quenching spectra during digestion.

### 3.10. Fluorescence Quenching

TP-FCR contained TP, and the free TP in the system exhibited a quenching effect on tryptophan and tyrosine, which were the main residues of the intrinsic fluorescence of proteins [63,65]. The fluorescence quenching spectrum of TP-FCR during digestion is shown in Figure 10A–F. All samples had a fluorescence emission peak and the fluorescence emission peaks of all samples decreased with the increase of digestion time. This was because the TP in TP-FCR was gradually released during digestion and more and more free TP interacted with digestive enzymes in the system. It has been reported that hydrogen bond-mediated amylose-polyphenol complexes were released during digestion and the interaction of polyphenols with other macromolecules such as starch, proteins, and lipids will affect the activity and release of the polyphenols [5,66,67]. Ortega et al. proposed that the lipids in cocoa liquid had a protective effect on cocoa polyphenols which enhanced the micellar and stability of the polyphenols [68]. As shown in Figure 10G, MCT played a dose-dependent effect on the quenching effect of TP in TP-FCR on digestive enzymes after 180 min of digestion. The more MCT was added, the stronger the quenching effect was. This can be attributed to the protective effect of MCT on TP in TP-FCR. In addition, the fluorescence peak value of TP-FCR decreased rapidly in the early stage of digestion and changed little during 60–180 min of digestion, whereas the fluorescence peak value of TP-FCR with MCT decreased continuously during 180 min of digestion. This suggested that MCT can regulate the release of TP in TP-FCR.

### 3.11. In Vitro Digestibility Properties

According to the above results, the addition of MCT can inhibit the digestion of TP-FCR, so the in vitro digestibility properties of TP-FCR were measured. The starch fraction and kinetic parameters obtained by fitting the starch hydrolysis rates with a first-order kinetics model were summarized in Table 3. R^2^ values ranged from 0.990 to 0.995, indicating that all fitted the first-order rate equation well [63,69]. The K value, HI, and pGI were negatively correlated with SDS and RS content and positively correlated with RDS content. As shown in Table 3, the RDS content, HI, and pGI of TP-FCR decreased with the increase of MCT, while the content of SDS and RS increased with increasing MCT. The results of in vitro digestion showed that the RDS content of cooked white rice was 74.28%. From Table 3, we observed that the RDS content of TP-FCR decreased to 64.43% compared to cooked white rice, and its content further decreased to 50.82% with the addition of 5% MCT. It demonstrated that MCT could inhibit the digestion of starch in TP-FCR, and the inhibitory effect was dose-dependent. Related studies have shown that the formation of starch-lipid complexes can reduce the sensitivity of starch to enzymes and affect the digestive properties of starch [6,59,70]. Simultaneously, TP can delay the hydrolysis of starch by binding with enzymes to denature digestive enzymes [5,23,47]. TP-FCR with MCT was highly resistant to digestion, and this resistance depended on multifaceted factors, including structural changes caused by the formation of starch-lipid complexes and the inhibitory effect of released polyphenols on amylolytic enzymes. On the one hand, the structural changes caused by the starch-MCT complexes and the attachment of MCT on the surface hindered the contact between enzymes and starch, thus inhibiting the digestion of starch. On the other hand, the protective effect of MCT on TP increased the inhibitory effect of TP on digestive enzymes in TP-FCR, thereby reducing the digestibility of TP-FCR. The above results indicated that the addition of MCT had a good improvement effect on the digestion of starch in TP-FCR.

## 4. Conclusions

This work suggested that the addition of MCT affected the adhesiveness, texture, color, and digestive properties of TP-FCR. Results revealed that MCT significantly improved the anti-digestibility of the TP-FCR. SEM, CLSM, DSC, XRD, FT-IR, and Raman spectroscopy demonstrated the important role of MCT addition in reducing the starch digestibility of TP-FCR from the perspective of multi-scale structural changes. Meanwhile, it can be concluded that the high anti-digestibility of TP-FCR with MCT can be attributed to structural changes caused by the formation of starch-lipid complexes and the inhibitory effect of released polyphenols on amylolytic enzymes. In addition, MCT reduced the tendency of TP-FCR grains to stick to each other and the bottom of the pot and improved the color of TP-FCR. This study provides a reference value for applying MCT as a quality improver to starchy foods with high glycemic index. However, in vivo animal studies and clinical trials are needed to further investigate the mechanisms involved in starch metabolism of TP-FCR with MCT.

## Figures and Tables

**Figure 1 foods-12-04366-f001:**
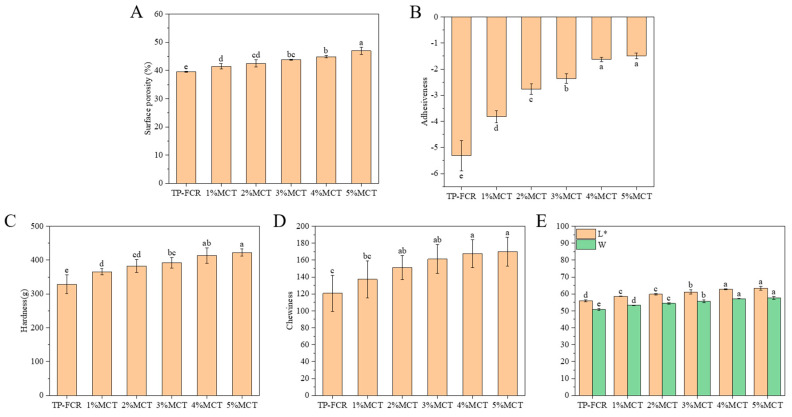
Porosity ratio (**A**), adhesiveness (**B**), hardness (**C**), chewiness (**D**), and color (**E**) of tea polyphenols-fortified cooked rice (TP-FCR) with/without MCT. Values in the same image with different letters are significantly different (*p* < 0.05).

**Figure 2 foods-12-04366-f002:**
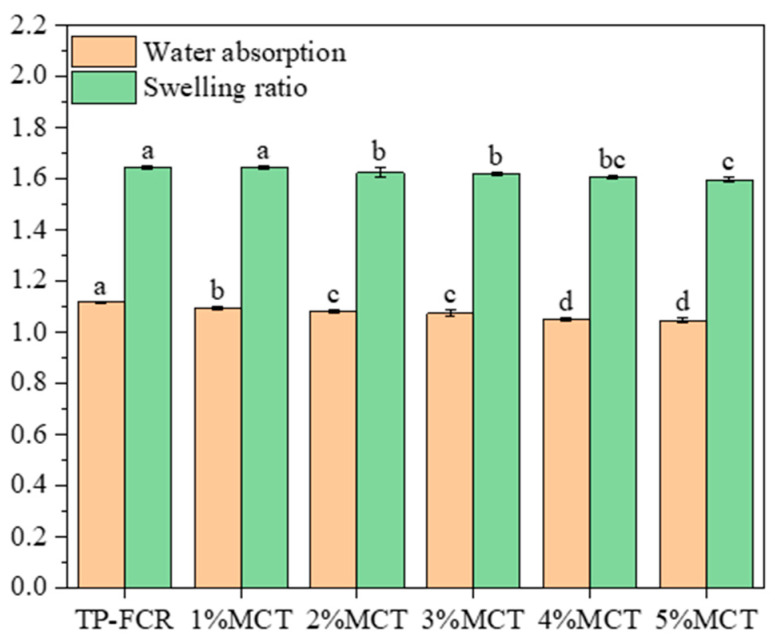
Water absorption and swelling ratio of tea polyphenols-fortified cooked rice (TP-FCR) with/without MCT. Values in the same image with different letters are significantly different (*p* < 0.05).

**Figure 3 foods-12-04366-f003:**
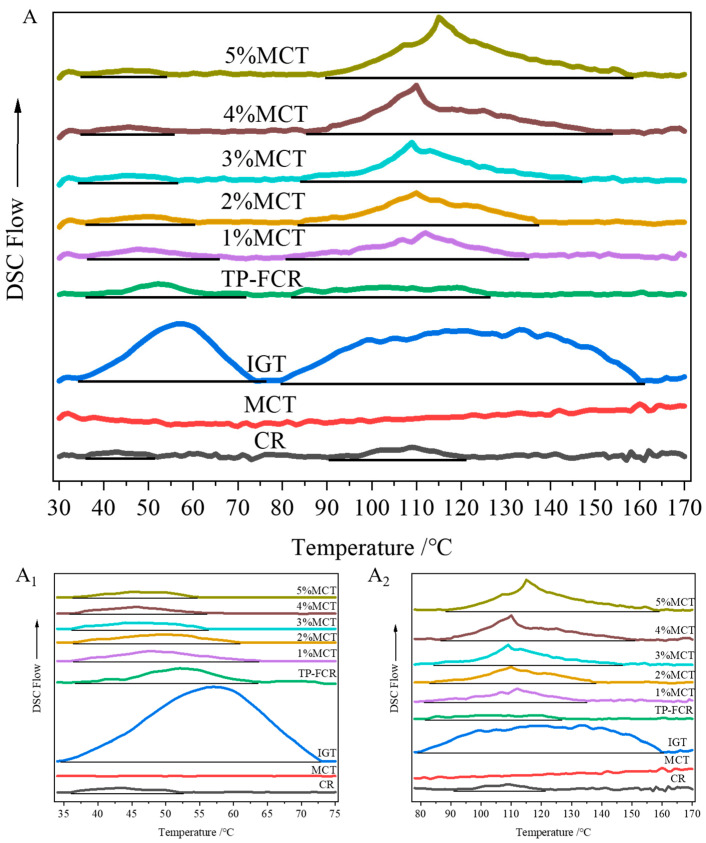
Thermodynamic properties (**A**,**A_1_**,**A_2_**) of tea polyphenols-fortified cooked rice (TP-FCR) with/without MCT.

**Figure 4 foods-12-04366-f004:**
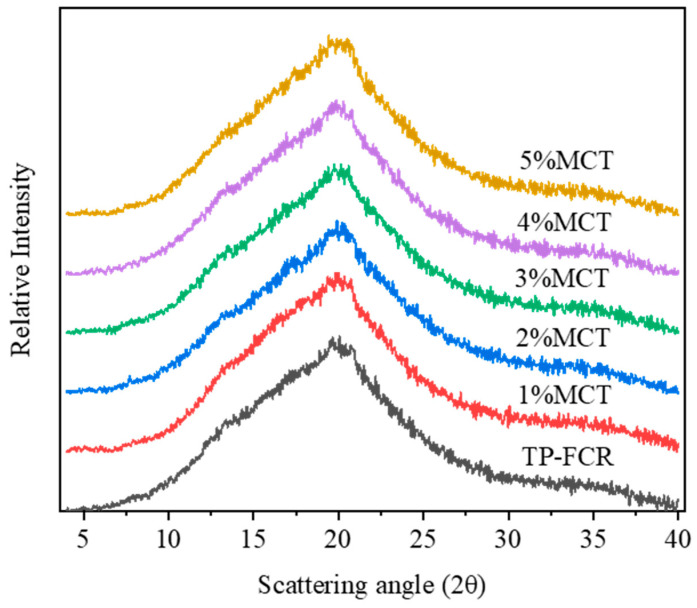
XRD spectra of tea polyphenols-fortified cooked rice (TP-FCR) with/without MCT.

**Figure 5 foods-12-04366-f005:**
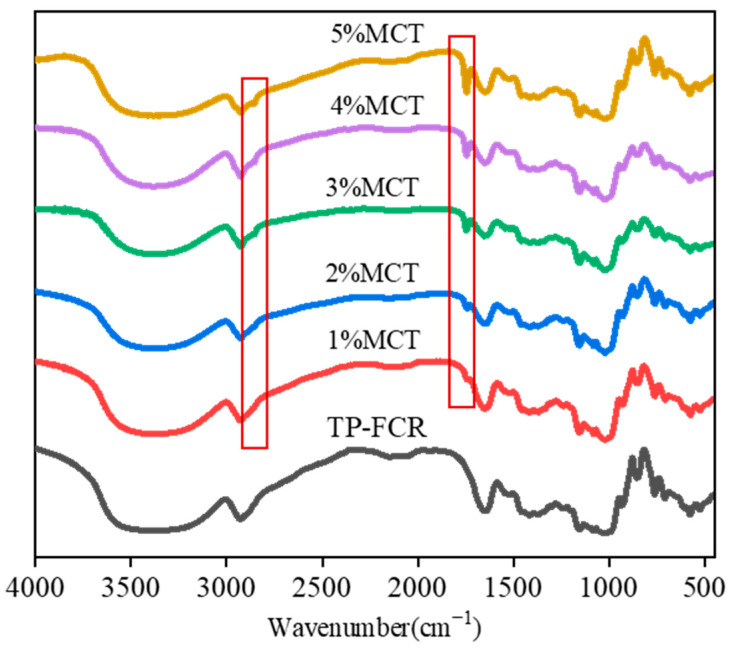
FTIR spectra of tea polyphenols-fortified cooked rice (TP-FCR) with/without MCT.

**Figure 6 foods-12-04366-f006:**
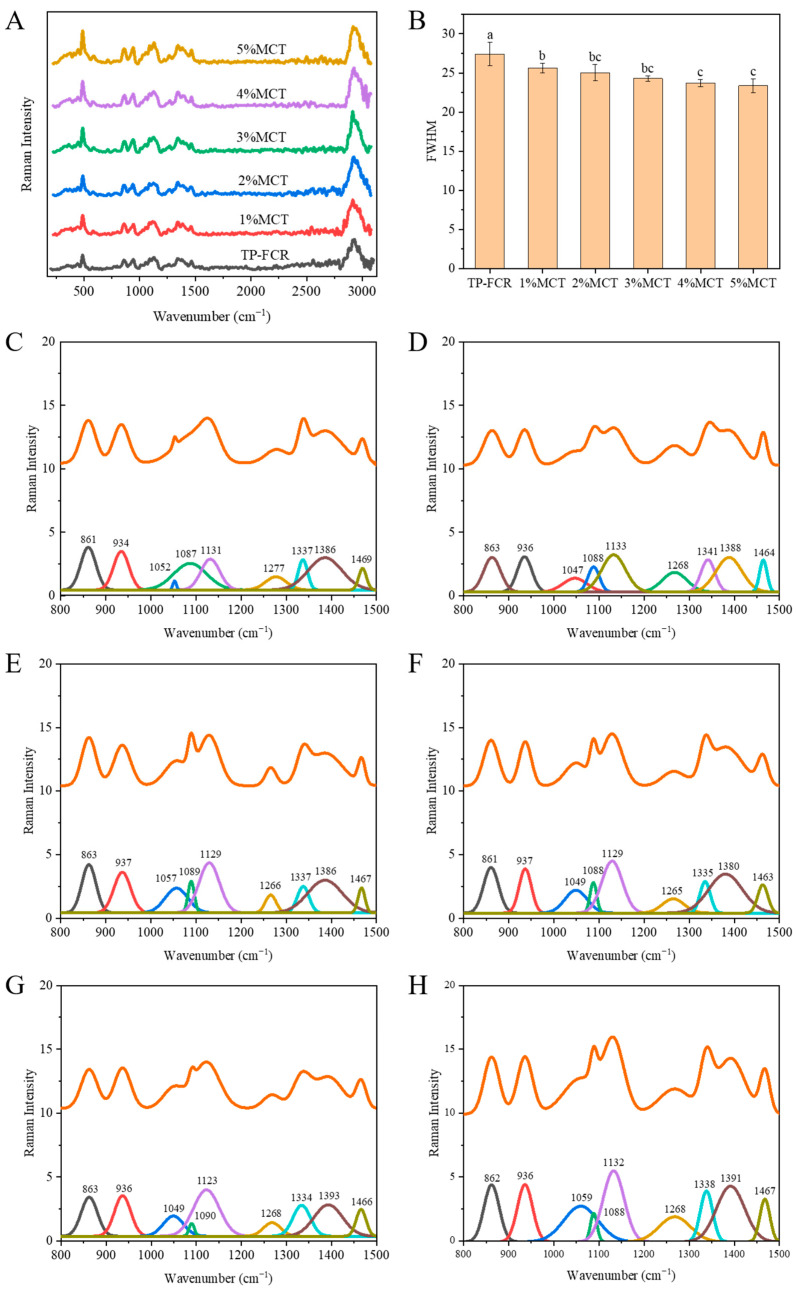
Raman spectra (**A**), FWHM at 480 cm^−1^ (**B**), and the characteristic bands of Raman spectra at 800–1500 cm^−1^ (**C**–**H**) of tea polyphenols-fortified cooked rice (TP-FCR) with/without MCT. Values in the same image with different letters are significantly different (*p* < 0.05).

**Figure 7 foods-12-04366-f007:**
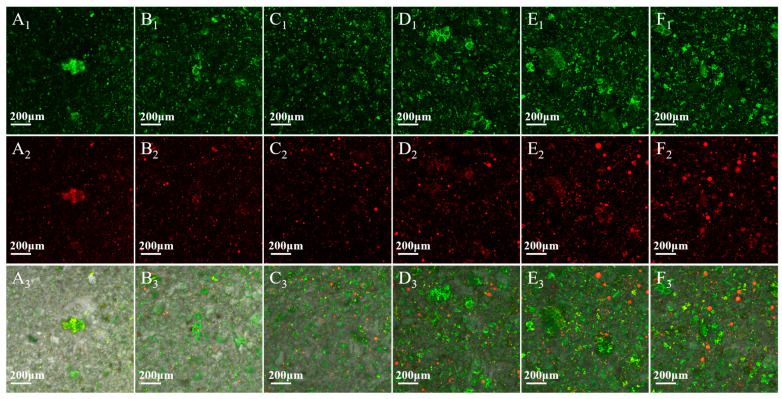
CLSM images of tea polyphenols-fortified cooked rice (TP-FCR) with/without MCT. (**A_1_**–**A_3_**): TP-FCR without MCT; (**B_1_**–**B_3_**): TP-FCR with 1% MCT; (**C_1_**–**C_3_**): TP-FCR with 2% MCT; (**D_1_**–**D_3_**): TP-FCR with 3% MCT; (**E_1_**–**E_3_**): TP-FCR with 4% MCT; (**F_1_**–**F_3_**): TP-FCR with 5% MCT. (**A_1_**–**F_1_**): samples stained by FITC; (**A_2_**–**F_2_**): samples stained by Nile Red; (**A_3_**–**F_3_**): Merged.

**Figure 8 foods-12-04366-f008:**
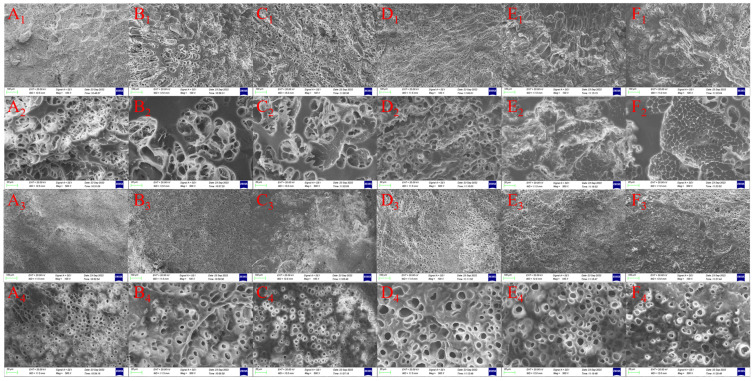
Surface and cross-section microstructure of tea polyphenols-fortified cooked rice (TP-FCR) with/without MCT. (**A_1_**–**A_4_**): TP-FCR without MCT; (**B_1_**–**B_4_**): TP-FCR with 1% MCT; (**C_1_**–**C_4_**): TP-FCR with 2% MCT; (**D_1_**–**D_4_**): TP-FCR with 3% MCT; (**E_1_**–**E_4_**): TP-FCR with 4% MCT; (**F_1_**–**F_4_**): TP-FCR with 5% MCT. ((**A_1_**–**F_1_**): 100× cross section microstructure; (**A_2_**–**F_2_**): 500× cross section microstructure; (**A_3_**–**F_3_**): 100× surface microstructure; (**A_4_**–**F_4_**): 500× surface microstructure.

**Figure 9 foods-12-04366-f009:**
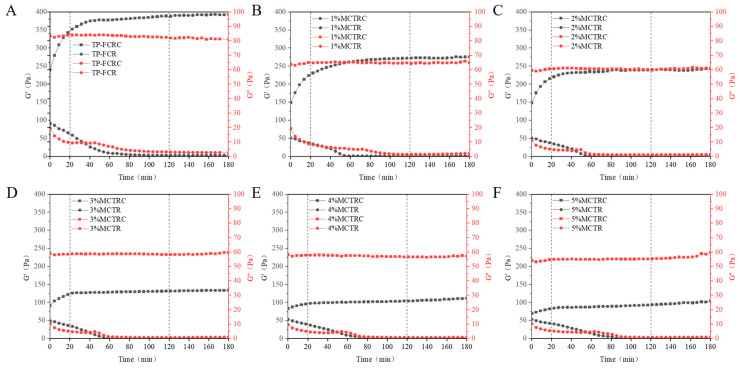
Dynamic viscoelasticity properties of tea polyphenols-fortified cooked rice (TP-FCR) with and without digestive enzymes (suffix “C”). TP-FCR without MCT (**A**); TP-FCR with 1% MCT (**B**); TP-FCR with 2% MCT (**C**); TP-FCR with 3% MCT (**D**); TP-FCR with 4% MCT (**E**); TP-FCR with 5% MCT (**F**).

**Figure 10 foods-12-04366-f010:**
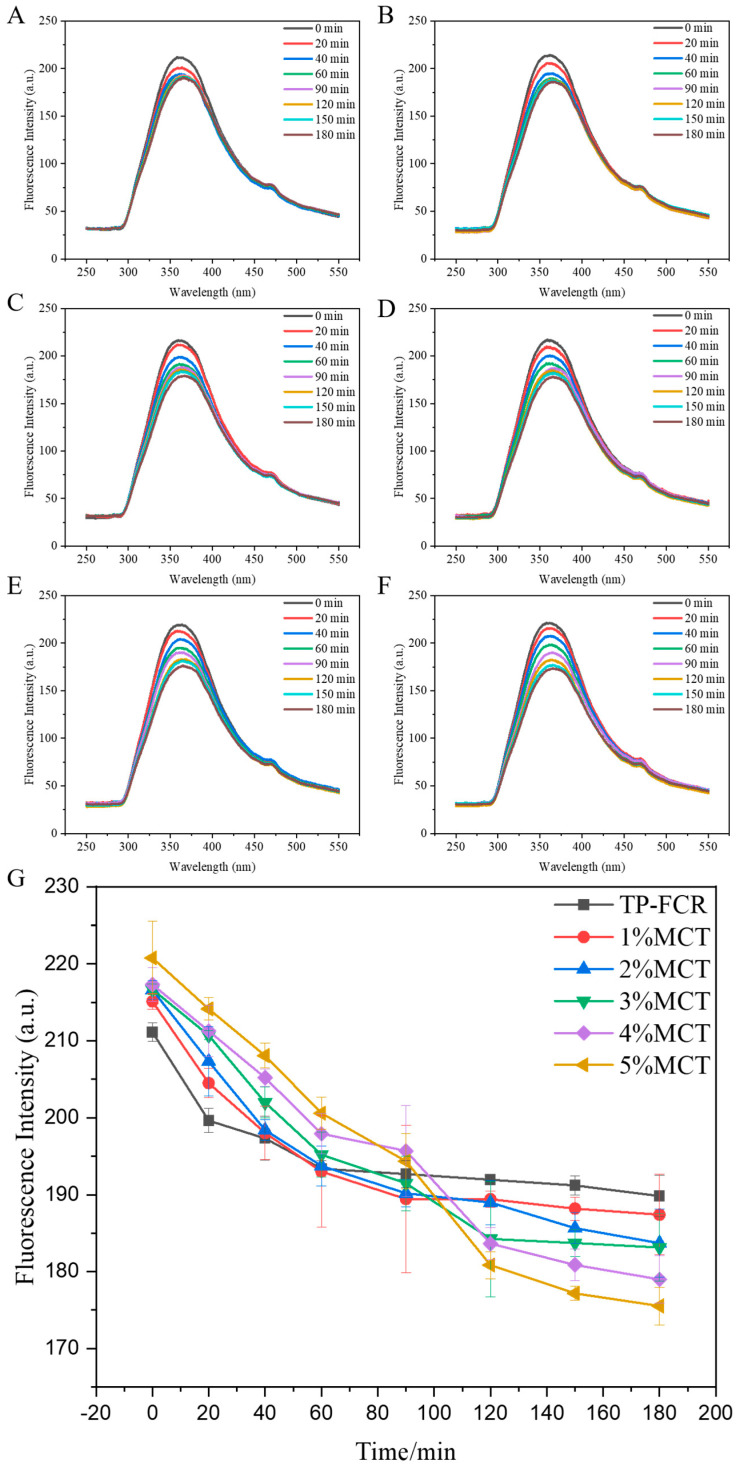
Fluorescence spectra during digestion (**A**–**F**) and the fluorescence peak value during digestion (**G**) of tea polyphenols-fortified cooked rice (TP-FCR) with/without MCT.

**Table 1 foods-12-04366-t001:** Thermodynamic properties of tea polyphenols-fortified cooked rice (TP-FCR) with/without MCT.

Samples	Peak I					Peak II				
	T_o_ (°C)	T_p_ (°C)	T_c_ (°C)	ΔH (J/g)	T_c_–T_o_ (°C)	T_o_ (°C)	T_p_ (°C)	T_c_ (°C)	ΔH (J/g)	T_c_–T_o_ (°C)
CR	36.36 ± 0.27 ^a^	43.50 ± 2.75 ^e^	52.58 ± 3.89 ^f^	0.34 ± 0.05 ^c^	16.22 ± 3.80 ^e^	91.41 ± 2.98 ^a^	103.65 ± 4.40 ^d^	120.11 ± 0.30 ^g^	1.41 ± 0.08 ^f^	28.70 ± 3.28 ^f^
MCT	ND	ND	ND	ND	ND	ND	ND	ND	ND	ND
IGT	35.25 ± 1.58 ^a^	57.48 ± 0.97 ^a^	73.93 ± 2.03 ^a^	11.94 ± 0.94 ^a^	38.68 ± 3.38 ^a^	79.92 ± 1.26 ^d^	121.22 ± 1.53 ^a^	159.07 ± 0.78 ^a^	25.41 ± 2.22 ^a^	79.15 ± 0.91 ^a^
TP-FCR	36.12 ± 0.83 ^a^	54.12 ± 0.96 ^b^	65.27 ± 0.97 ^b^	1.16 ± 0.09 ^b^	29.15 ± 0.47 ^b^	81.18 ± 2.36 ^d^	107.35 ± 0.51 ^c^	127.75 ± 4.08 ^f^	2.08 ± 0.31 ^f^	46.58 ± 1.75 ^e^
1%MCT	35.72 ± 0.60 ^a^	53.63 ± 0.74 ^bc^	63.40 ± 1.03 ^bc^	0.84 ± 0.01 ^bc^	27.68 ± 1.28 ^b^	83.17 ± 1.58 ^d^	107.91 ± 0.68 ^c^	133.27 ± 1.97 ^e^	3.85 ± 0.73 ^e^	50.10 ± 3.51 ^e^
2%MCT	35.69 ± 0.39 ^a^	53.55 ± 0.64 ^bc^	61.62 ± 1.40 ^cd^	0.73 ± 0.02 ^bc^	25.94 ± 1.79 ^bc^	85.86 ± 2.19 ^cd^	108.57 ± 1.00 ^c^	141.38 ± 3.77 ^d^	6.09 ± 0.27 ^d^	55.52 ± 2.26 ^d^
3%MCT	35.62 ± 0.38 ^a^	53.18 ± 1.11 ^bc^	58.76 ± 0.98 ^de^	0.62 ± 0.01 ^bc^	23.14 ± 1.32 ^cd^	86.17 ± 1.47 ^bc^	108.87 ± 0.75 ^c^	147.37 ± 4.73 ^cd^	7.22 ± 0.14 ^cd^	61.2 ± 4.88 ^c^
4%MCT	36.16 ± 0.30 ^a^	50.37 ± 4.18 ^cd^	57.04 ± 1.49 ^e^	0.56 ± 0.01 ^bc^	20.88 ± 1.64 ^d^	88.74 ± 2.63 ^ab^	109.79 ± 0.81 ^c^	153.14 ± 3.04 ^c^	8.63 ± 0.35 ^c^	64.41 ± 0.56 ^bc^
5%MCT	36.56 ± 0.17 ^a^	48.68 ± 0.40 ^d^	56.64 ± 1.27 ^e^	0.46 ± 0.01 ^c^	20.08 ± 1.21 ^d^	88.85 ± 0.26 ^ab^	113.90 ± 1.08 ^b^	157.10 ± 1.50 ^b^	10.69 ± 0.58 ^b^	68.25 ± 1.68 ^b^

Different letters in the same column showed significant differences (*p* < 0.05). T_o_, the starting temperature; T_p_, the peak temperature; T_c_, the termination temperature; ΔH, the enthalpy value; T_c_–T_o_, the heat absorption peaks range.

**Table 2 foods-12-04366-t002:** Relative crystallinity, IR ratios, and Raman peak area of tea polyphenols-fortified cooked rice (TP-FCR) with/without MCT.

Samples	Relative Crystallinity (%)	IR Ratio		Raman Peak Area	
		1045/1020 (cm^−1^)	1020/995 (cm^−1^)	1130 (cm^−1^)	1340 (cm^−1^)
TP-FCR	0.87 ± 0.07 ^d^	0.642 ± 0.018 ^d^	1.168 ± 0.020 ^a^	164.60 ± 10.12 ^d^	52.57 ± 5.80 ^d^
1%MCT	2.48 ± 0.13 ^c^	0.658 ± 0.013 ^cd^	1.145 ± 0.007 ^ab^	208.83 ± 14.69 ^c^	67.94 ± 3.67 ^c^
2%MCT	2.62 ± 0.12 ^c^	0.668 ± 0.002 ^bc^	1.151 ± 0.013 ^ab^	230.95 ± 15.35 ^bc^	69.10 ± 3.05 ^c^
3%MCT	2.96 ± 0.14 ^b^	0.674 ± 0.005 ^bc^	1.139 ± 0.006 ^ab^	258.84 ± 15.09 ^b^	85.03 ± 5.66 ^b^
4%MCT	4.01 ± 0.14 ^a^	0.683 ± 0.007 ^b^	1.121 ± 0.015 ^bc^	329.95 ± 29.29 ^a^	88.91 ± 0.58 ^b^
5%MCT	4.26 ± 0.20 ^a^	0.711 ± 0.002 ^a^	1.099 ± 0.026 ^c^	355.29 ± 23.44 ^a^	107.56 ± 1.25 ^a^

Different letters in the same column showed significant differences (*p* < 0.05).

**Table 3 foods-12-04366-t003:** Starch fractions and kinetic parameters of tea polyphenols-fortified cooked rice (TP-FCR) with/without MCT.

Samples	Starch Fraction			Kinetic Parameters			
	RDS (%)	SDS (%)	RS (%)	K (×10^−2^/min)	R^2^	HI (%)	pGI
TP-FCR	64.43 ± 1.74 ^a^	11.21 ± 1.00 ^e^	24.26 ± 0.72 ^e^	9.54 ± 0.50 ^a^	0.995	71.79	70.08
1%MCT	60.74 ± 1.80 ^b^	12.94 ± 0.09 ^d^	26.32 ± 0.98 ^d^	8.70 ± 0.17 ^b^	0.994	69.58	68.18
2%MCT	57.37 ± 0.79 ^c^	13.63 ± 0.22 ^cd^	28.99 ± 0.08 ^c^	8.25 ± 0.07 ^c^	0.990	66.80	65.78
3%MCT	55.05 ± 2.52 ^cd^	13.89 ± 0.11 ^bc^	31.07 ± 0.04 ^b^	8.01 ± 0.04 ^c^	0.990	64.76	64.02
4%MCT	52.39 ± 1.41 ^de^	15.29 ± 0.26 ^a^	32.32 ± 1.79 ^b^	7.48 ± 0.06 ^d^	0.991	63.13	62.62
5%MCT	50.82 ± 0.88 ^e^	14.68 ± 0.18 ^ab^	34.52 ± 1.77 ^a^	7.44 ± 0.08 ^d^	0.994	61.13	60.89

Different letters in the same column showed significant differences (*p* < 0.05).

## Data Availability

Data is contained within the article and Appendix A.

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
