# Peer review of "Effect of Medium Chain Triglycerides on the Digestion and Quality Characteristics of Tea Polyphenols-Fortified Cooked Rice"

_foods, 2023, doi:10.3390/foods12234366_

Round 1
Reviewer 1 Report
Comments and Suggestions for Authors
The gycemic index of foods in general and rice specifically is an important research topic. Only a few comments and suggestions are provided for the authors below.
In the future please include line numbers in your documents. It makes it easier for the reviewers.
What sensory quality type of rice was used. Only the brand name is provided. The sensory quality is associated with the amylose content. It is the amylose in rice primarily that complexes with lipids. The rice name includes pearl rice. Thus, an assumption can be made that this is a low amylose type of rice. But, this needs to be documented in the rice description.
A reference for Image j1.48u software is needed.
Was there a true control run in this study? No tea extract and no MCT? Why not?
The methods indicate that the cooked rice was stored in plastic bags to stop moisture loss. Some plastic allows mosture transmission. The MVTR (Moisture Vapor Transmission Rate) of the plaastid needs to be reported.
The following is confusing - "A completely grain of TP-FCR was placed in the center of the test bench."
What rice samples specifically were used for the textural analysis? Rice texture is different within a rice cooker - edges versus the center. Rice texture is different based on storage time after cooking temperature of cooked rice during storage. Storage is defined as the time after cooking and collecting the texture data.
"then mixed well with the pre-activated digestive enzyme solution (PPAAAG=2)" Explain pre-activated.
The statistical methods are lacking. No mention of ANOVA is reported, yet Duncans Multiple Range test is described. The former must be run before the later. Why wasn't this done? This puts most of the results into question.
Hardness and chewiness are discussed in the results section but defined in the methods section. This needs to be done. Also, the authors need to discuss whether or not these instrumental measurements are correlated with human sensory data.
The results section states the authors know why they found the results that they did. But, only one reference is provided to support their conclusions. There are many other related studies.
Nothing in the methods section indicates how shininess or luster was measured. Also, the authors need to be clear when they are discussing results from sensory studies versus instrumental studies. They don't always correlate highly. "Shiny cooked rice was more appetizing and more popular [17]. It was observed from Figure 1E that the L* and W were increased with the addition of MCT. And the more MCT was added, the greater the L* and W were, which indicated that the whiteness and brightness of TP-FCR was dose-dependent on MCT. This suggested that MCT can enhance the luster of TP-FCR thereby improving its color."
The conclusions section needs to be rewritten. The following words need to be defined or another words used - What does it mean to improve the quality and digestive properties of rice? What is the edible value of rice?
The results in this study don't support the following - ", but also for reducing dietary glycemic load and reducing the risk of dietary related diseases." The results only indicate that this rice food Might or May do these things. Human based controlled studies are needed to verify these results. The limitations of this study must be discussed. Instrumental data isn't the same has clinical trials.
How do the authors propose that this rice would be available for consumers? Will they add the
Abstract - "The findings suggested that MCT incorporation could be a conveniently promising strategy to apply in cooked rice production to achieve high-eating quality and low-glycemic cooked rice without a safety evaluation." Be very careful stating high eating quality. Different consumers prefer different quality types of rice. This study only evaluated what appears to be low amylose rice. Much of the world prefers higher amylose rice. Plus, this study only used instrumental data, no human sensory data was collected. Thus, it isn't correct to say this study found high eating quality rice after the treatments. Also, what is meant by "without a safety evaluation."
Comments on the Quality of English Language
The English in some areas is quite good while in others is a bit hard to follow.
Author Response
Reviewer #1:
The glycemic index of foods in general and rice specifically is an important research topic. Only a few comments and suggestions are provided for the authors below.
Comment 1: In the future, please include line numbers in your documents. It makes it easier for the reviewers.
Response: Thank you for your comment. The line numbers have been added in our manuscript.
Comment 2: What sensory quality type of rice was used. Only the brand name is provided. The sensory quality is associated with the amylose content. It is the amylose in rice primarily that complexes with lipids. The rice name includes pearl rice. Thus, an assumption can be made that this is a low amylose type of rice. But this needs to be documented in the rice description.
Response: Thank you for your comment. The amylose content of rice has been added and marked red in the manuscript (Line 90).
Comment 3: A reference for Image j1.48u software is needed.
Response: The reference has been added and marked red in the manuscript (Line 120). Thanks.
Comment 4: Was there a true control run in this study? No tea extract and no MCT? Why not?
Response: The control sample in this study was tea polyphenols - fortified cooked rice without MCT. Because our research group had previously studied the effects of instant green tea on cooked rice which was published in Food & Function (Effects of tea products on in vitro starch digestibility and eating quality of cooked rice using domestic cooking method). As a control, the properties of white cooked rice without tea extract and MCT have been comprehensively studied in that paper. This work aimed to investigate the effects of MCT on TP-FCR, so white cooked rice was not studied. Thanks.
Comment 5: The methods indicate that the cooked rice was stored in plastic bags to stop moisture loss. Some plastic allows moisture transmission. The MVTR (Moisture Vapor Transmission Rate) of the plastic needs to be reported.
Response: Thank you for your comment. The MVTR (Moisture Vapor Transmission Rate) of the plastic bags has been added and marked red in the manuscript (Line 109-110).
Comment 6: The following is confusing - "A completely grain of TP-FCR was placed in the center of the test bench."
Response: Thank you for your comment. The sentence has been rewritten and marked red in the manuscript (Line 124-126).
Comment 7: What rice samples specifically were used for the textural analysis? Rice texture is different within a rice cooker - edges versus the center. Rice texture is different based on storage time after cooking temperature of cooked rice during storage. Storage is defined as the time after cooking and collecting the texture data.
Response: The corresponding details have been added and marked red in the manuscript (Line 124-126). Thank you.
Comment 8: "then mixed well with the pre-activated digestive enzyme solution (PPA: AAG=20:1)" Explain pre-activated.
Response: The explanation of “pre-activated” has been supplemented and marked red in the manuscript (Line 195-198).Thanks.
Comment 9: The statistical methods are lacking. No mention of ANOVA is reported, yet Duncans Multiple Range test is described. The former must be run before the later. Why wasn't this done? This puts most of the results into question.
Response: Thank you for your comment. The statistical methods have been redescribed and marked red in the manuscript (Line 245-246).
Comment 10: Hardness and chewiness are discussed in the results section but defined in the methods section. This needs to be done. Also, the authors need to discuss whether or not these instrumental measurements are correlated with human sensory data.
Response: Thank you for your comment, the corresponding content has been added and marked red in the manuscripts (Line 128-132 and Line 263-270). And sensory evaluation of TP-FCR with/without MCT was added in the supplementary material.
Comment 11: The results section states the authors know why they found the results that they did. But, only one reference is provided to support their conclusions. There are many other related studies.
Response: More references have been added to the results section and marked red in the manuscript. Thanks.
Comment 12: Nothing in the methods section indicates how shininess or luster was measured. Also, the authors need to be clear when they are discussing results from sensory studies versus instrumental studies. They don't always correlate highly. "Shiny cooked rice was more appetizing and more popular [17]. It was observed from Figure 1E that the L* and W were increased with the addition of MCT. And the more MCT was added, the greater the L* and W were, which indicated that the whiteness and brightness of TP-FCR was dose-dependent on MCT. This suggested that MCT can enhance the luster of TP-FCR thereby improving its color."
Response: This section has been rewritten and marked red in the manuscripts (Line 136-137, 279-282 and Line 284-285). Thank you.
Comment 13: The conclusions section needs to be rewritten. The following words need to be defined or another words used - What does it mean to improve the quality and digestive properties of rice? What is the edible value of rice?
Response: Thank you for your comment. The conclusion section has been rewritten and marked red in the manuscript (Line 561-573).
Comment 14: The results in this study don't support the following - ", but also for reducing dietary glycemic load and reducing the risk of dietary related diseases." The results only indicate that this rice food Might or May do these things. Human based controlled studies are needed to verify these results. The limitations of this study must be discussed. Instrumental data isn't the same has clinical trials.
Response: These are great comments and we appreciate the critique. It has been rewritten and marked red in the manuscript (Line 570-573).
Comment 15: How do the authors propose that this rice would be available for consumers? Will they add the
Response: Thank you for your comment. Tea polyphenols-fortified cooked rice (TP-FCR) is a traditional staple that is popular among consumers for its unique flavor and nutritional value. Nowadays, medium chain triglycerides with good physicochemical and functional properties have been increasingly applied for fortifying food products. MCT as a food ingredient is easily purchased and can be added in a convenient way like adding water to rice. Therefore, TP-FCR with MCT will be available for consumers.
Comment 16: Abstract - "The findings suggested that MCT incorporation could be a conveniently promising strategy to apply in cooked rice production to achieve high-eating quality and low-glycemic cooked rice without a safety evaluation." Be very careful stating high eating quality. Different consumers prefer different quality types of rice. This study only evaluated what appears to be low amylose rice. Much of the world prefers higher amylose rice. Plus, this study only used instrumental data, no human sensory data was collected. Thus, it isn't correct to say this study found high eating quality rice after the treatments. Also, what is meant by "without a safety evaluation." Response: Thank you for your comment. It has been revised and marked red in the manuscript (Line 25-26). And the sensory evaluation of TP-FCR with different addition of MCT was added in the supplementary material. The results showed that the addition of MCT conferred better sensory quality and higher overall acceptability of TP-FCR. The sensory evaluation results were consistent with the instrumental measurements results.
Comment 17: The English in some areas is quite good while in others is a bit hard to follow.
Response: Thank you for your comment. We have tried our best to polish the language in the manuscript, and the revised portions were highlighted in red.

Reviewer 2 Report
Comments and Suggestions for Authors
The subject of the publication is interesting. In some places, the English language should be improved to be more understandable or the issue presented should be better described (details below). Above all, however, there is a lack of significant information determining the scientific quality of the research and the adequacy of the description of the results. Namely, an important part of the research material was not characterized: instant green tea (what did the preparation consist of? how was it made? did it contain a carrier? saccharides?) and MCT (what was the composition of fatty acids? what was the purity of triacylglycerols?). This is the basic material used for research and knowledge of its composition is key to interpreting the results.
Other notes:
methods
water absorption rate: not % from the formula; besides, it describes not only the water absorbtion, but also the other ingredients used to prepare the sample
swelling ratio: not %; if 10g of sample was taken after cooking, it should be simply V2 x 10
results
page 6: MCT has good properties such as extensibility, solubility (in water used in experiments?), emulsion stability? were these pure triacylglycerols?
page 8: peak I and peak II - not sufficiently explained in the text and description of the figure
page 9: table 1 - unreadable values in the table layout used
page 19 in vitro digestibility: correlations should be made between starch fractions and other parameters declared in the discussion of the results as changing analogously to these fractions
Please read the text carefully in terms of the quality of the English language. Examples of questionable places (page/paragraph from the top):
1/1: that boils rice with tea
2/4: emusifying
while - not needed or one sentence
3/2: filtered it quickly
kept it warm
3/5: completely
6/3: give with better
7/1: It can be the fact
14/1: that the more
Author Response
Reviewer #2:
Comment 1: The subject of the publication is interesting. In some places, the English language should be improved to be more understandable or the issue presented should be better described (details below). Above all, however, there is a lack of significant information determining the scientific quality of the research and the adequacy of the description of the results. Namely, an important part of the research material was not characterized: instant green tea (what did the preparation consist of? how was it made? did it contain a carrier? saccharides?) and MCT (what was the composition of fatty acids? what was the purity of triacylglycerols?). This is the basic material used for research and knowledge of its composition is key to interpreting the results.
Response: Thank you for your comment. The English language of this paper has been carefully revised. And more details about the research material have been supplemented and marked red in the manuscript (Line 92-94 and Line 95-98).
Methods
Comment 2: water absorption rate: not % from the formula; besides, it describes not only the water absorption, but also the other ingredients used to prepare the sample.
Response: Thank you for your comment. The formula for the water absorption rate has been revised and marked red in the manuscript (Line 143).
Comment 3: swelling ratio: not %; if 10g of sample was taken after cooking, it should be simply V2 x 10.
Response: Thank you for your comment. The formula for the swelling ratio has been revised and marked red in the manuscript (Line 151). W in this study is the weight of 100 g of rice after cooking instead of 100 g, so the formula for the swelling ratio was described as follows:
Results
Comment 4: page 6: MCT has good properties such as extensibility, solubility (in water used in experiments?), emulsion stability? were these pure triacylglycerols?
Response: Thank you for your comment. It has been redescribed and the purity of MCT is 100% (Line 256-258).
Comment 5: page 8: peak I and peak II - not sufficiently explained in the text and description of the figure.
Response: More descriptions and explanations of Peak I and peak II have been added and marked red in the manuscript (Line 316-321). Thanks.
Comment 6: page 9: table 1 - unreadable values in the table layout used
Response: This table has been revised. Thank you.
Comment 7: page 19 in vitro digestibility: correlations should be made between starch fractions and other parameters declared in the discussion of the results as changing analogously to these fractions.
Response: Thank you for your comment. It has been added and marked red in the manuscript (Line 533-536).
Comments on the Quality of English Language
Comment 8: Please read the text carefully in terms of the quality of the English language. Examples of questionable places (page/paragraph from the top):
1/1: that boils rice with tea
2/4: emusifying
while - not needed or one sentence
3/2: filtered it quickly
kept it warm
3/5: completely
6/3: give with better
7/1: It can be the fact
14/1: that the more
Response: Thank you for your comment. We have revised the above questionable places and tried our best to polish the language in the manuscript. The revised portions have been marked red in the manuscript.
